# CRISPR knockouts reveal an endogenous role for ancient neuropeptides in regulating developmental timing in a sea anemone

**Nagayasu Nakanishi[1]\*, Mark Q Martindale[2]**

[1]Department of Biological Sciences, University of Arkansas, Fayetteville, United States; [2]Whitney Laboratory for Marine Bioscience, University of Florida, St. Augustine, United States

**Abstract** Neuropeptides are evolutionarily ancient peptide hormones of the nervous and neuroendocrine systems, and are thought to have regulated metamorphosis in early animal ancestors. In particular, the deeply conserved Wamide family of neuropeptides—shared across Bilateria (e.g. insects and worms) and its sister group Cnidaria (e.g. jellyfishes and corals)—has been implicated in mediating life-cycle transitions, yet their endogenous roles remain poorly understood. By using CRISPR-Cas9-mediated reverse genetics, we show that cnidarian Wamide—referred to as GLWamide—regulates the timing of life cycle transition in the sea anemone cnidarian *Nematostella vectensis*. We find that mutant planula larvae lacking GLWamides transform into morphologically normal polyps at a rate slower than that of the wildtype control larvae. Treatment of GLWamide null mutant larvae with synthetic GLWamide peptides is sufficient to restore a normal rate of metamorphosis. These results demonstrate that GLWamide plays a dispensable, modulatory role in accelerating metamorphosis in a sea anemone.

DOI: https://doi.org/10.7554/eLife.39742.001

\*For correspondence:
nnakanis@uark.edu

**Competing interests:** The authors declare that no competing interests exist.

## Introduction

Neuropeptides are evolutionarily ancient hormones—secreted, diffusible signaling-molecules—that are expressed in many cells of the nervous and neuroendocrine systems of Bilateria (e.g. vertebrates, insects and worms) and its sister group Cnidaria (e.g. jellyfishes, corals and sea anemones) (*Grimmelikhuijzen and Hauser, 2012*; *Grimmelikhuijzen et al., 2002*; *Hartenstein, 2006*; *Walker et al., 2009*). They are short polypeptides generated from larger precursor proteins through proteolytic cleavage, and are stored in membrane-bound secretory vesicles that are distributed throughout the neuron. For some, peptide α-amidation occurs via a post-translational, enzymatic conversion of a C-terminal glycine into an amide group, and is often critical for biological activity (reviewed in [*Eipper et al., 1992*]). Intercellular signaling occurs when neuropeptides are released from the secretory vesicles into the extracellular space and received by receptors of target cells. Neuropeptides have been implicated in regulating a wide range of biological processes—from behavior, physiology, and development—in animals with nervous systems (summarized in [*Strand, 1999*]). Given that neuropeptides were early neurotransmitters in the evolution of nervous systems, testing hypotheses about the conserved roles of neuropeptides in extant animals is directly relevant to understanding how the primitive nervous system may have functioned.

Comparative genomic evidence suggests that several peptide hormones are shared across Bilateria and Cnidaria, namely an insulin-related peptide, glycoprotein hormone, prokineticin, and amidated peptides with carboxy-terminal sequences Wamide, RFamide and RYamide (*Jékely, 2013*).

Among them, only amidated peptides have been shown to occur in neurons of both bilaterians and cnidarians (e.g. (*Conzelmann and Jékely, 2012*; *Conzelmann et al., 2013*; *Grimmelikhuijzen, 1985*; *Koizumi et al., 2004*; *Schmich et al., 1998a*); see also (*Walker et al., 2009*) for a review of literature on RFamide), and thus are here considered *bona fide* conserved neuropeptides. The precursor protein of these neuropeptides typically encodes multiple copies of immature neuropeptides (e.g. (*Conzelmann and Jékely, 2012*; *Darmer et al., 1991*; *Leviev and Grimmelikhuijzen, 1995*), some of which can differ in amino acid sequence; this indicates that an immature neuropeptide sequence can duplicate and diverge within a gene. Paralogous neuropeptides can thus be generated from a single gene, or from different genes that emerged by gene duplication (i.e. paralogous genes). Neuropeptides can also be lost during evolution; for example, deuterostome bilaterians (e.g. chordates and echinoderms) lack Wamide precursor orthologs (*Conzelmann et al., 2013*), and the RYamide-encoding gene appears absent from the genome of anthozoan cnidarians (sea anemones and corals; e.g. (*Anctil, 2009*).

The ancestral function of the conserved neuropeptides is enigmatic, but a few lines of evidence suggest that Wamide may have conserved roles in metamorphosis regulation across Bilateria and Cnidaria (*Conzelmann et al., 2013*; *Schoofs and Beets, 2013*). First, Wamide is expressed in larval nervous systems in bilaterians (e.g. (*Conzelmann et al., 2013*; *Hua et al., 1999*) and cnidarians (e.g. [*Leitz and Lay, 1995*]). Second, exogenous Wamides have effects on molecular, behavioral and/or morphogenetic processes underlying life cycle transition across animals. For instance, Wamide neuropeptides (also known as allatostatin-B, prothoracicostatic peptides, myoinhibitory peptides (MIP), and GLWamide) can inhibit the synthesis of juvenile hormone in crickets (*Lorenz et al., 1995*) and that of ecdysteroids (the molting hormone) in a silkworm (*Hua et al., 1999*). Also, they can induce larval settlement of trochophore larvae in annelids (*Conzelmann et al., 2013*), and metamorphosis of planula larvae into polyps in hydrozoan and scleractinian (hard coral) cnidarians (*Erwin and Szmant, 2010*; *Iwao et al., 2002*; *Leitz et al., 1994*; *Schmich et al., 1998b*). Third, Wamide signaling appears to be necessary for life cycle transition across Bilateria and Cnidaria. The evidence comes from the annelid bilaterian *Platynereis dumerilii*, where morpholino-mediated knockdown of MIP receptor expression blocked MIP-induced larval settlement behavior (*Conzelmann et al., 2013*); but see Discussion), and the hydrozoan cnidarian *Hydractinia echinate*, where RNAi-mediated knockdown of GLWamide precursor expression can decrease rates of metamorphosis, and synthetic GLWamide could rescue metamorphosis-less phenotype that resulted from pharmacological inhibition of neuropeptide amidation (*Plickert et al., 2003*). However, reduced rates of metamorphosis in GLWamide-deficient hydrozoan larvae were not consistently observed across independent experiments despite a near-complete loss of endogenous GLWamide precursor transcripts. It was therefore proposed that quantitatively small amounts of GLWamide might be sufficient for larvae to remain competent to metamorphosis induction. This hypothesis has yet to be directly confirmed by gene knockout approaches.

Here we used CRISPR-Cas9-mediated mutagenesis to generate knockout mutant lines for a Wamide precursor gene in the sea anemone cnidarian *Nematostella vectensis,* and examined the development of knockout mutants relative to the wildtype. In contrast to the hypothesis that Wamide is necessary for life cycle transition in *H. echinate*, we report that Wamide knockout mutant larvae can transform into morphologically normal polyps in *N. vectensis*. However, knockout mutant larvae undergo metamorphosis at slower rates than the wildtype control larvae, and this developmental phenotype could be rescued by treatment of the mutant larvae with synthetic GLWamide peptides. These results demonstrate that Wamide has a dispensable, modulatory role as an accelerator of metamorphosis in *N. vectensis*.

## Results

The genome of the sea anemone *Nematostella vectensis* (*Putnam et al., 2007*) encodes one Wamide (GLWamide, MH939200) precursor homolog (Nv126270) (*Anctil, 2009*). The GLWamide precursor gene was predicted to contain a single exon based on in silico gene prediction (http://genome.jgi.doe.gov/cgi-bin/dispGeneModel?db=Nemve1&id=126270); however, comparison of the *N. vectensis* genome and cDNA sequences instead shows that the GLWamide gene consists of two exons (*Figure 1—figure supplement 1A*). The analysis of putative endoproteolytic cleavage sites (i.e. acidic and basic amino acid residues; (*Darmer et al., 1991*; *Leviev and Grimmelikhuijzen,*

*1995*) and C-terminal amidation sites (i.e. C-terminal Gly residue) in the predicted precursor protein sequence suggests that the GLWamide gene encodes different copy numbers of nine distinct GLWamide peptides that vary in N-terminal sequence (*Figure 1—figure supplement 1B,C*). We note that previously documented NvLWamide-like gene (Nv242283; (*Havrilak et al., 2017*) is predicted to generate and release QCPP<u>GLW</u>GC peptides, but not GLWamides, because of the lack of the canonical amidation signal, glycine, at the C-terminus (see *Figure 5—figure supplement 1* for experimental validation).

We first characterized the spatiotemporal expression patterns of GLWamide during planula development in *N. vectensis*. We used in situ hybridization and immunostaining to detect GLWamide precursor transcripts and GLWamide peptides, respectively. Consistent with previously published gene/peptide expression data (*Nakanishi et al., 2012*; *Watanabe et al., 2014*), GLWamide precursor transcripts were first detected in ectodermal sensory cells in the outer epithelium and in the pharynx during early-mid planula development (arrowheads in *Figure 1A,B*; inset in A) when a bundle of long cilia known as the apical tuft develops at the aboral pole. At the mid-planula stage, GLWamide transcripts appear in a subset of endodermal epithelial cells in the body column (arrowheads in *Figure 1D*). Interestingly, GLWamide-positive sensory cells in the pharynx were often asymmetrically distributed along the directive axis, oriented perpendicular to the oral-aboral axis (inset in *Figure 1D*; [*Watanabe et al., 2014*]).

Immunostaining revealed GLWamide peptide expression in a number of neuronal processes of the ectodermal and endodermal nervous systems at the mid-late planula stage (*Figure 2A,B*). Ectodermal sensory cells of the body column extend basal processes aborally or laterally, but not toward the oral direction (*Figure 2C,D*), while GLWamide-positive pharyngeal sensory cells extend neurites in any direction (*Figure 2E,F*). In the endoderm, GLWamide expression is detected in neurons that form longitudinal neurite bundles (*Figure 2G,H*). These results demonstrate that, prior to transformation into a polyp through formation of oral tentacles, GLWamides are expressed in neurons and their neurites that constitute ectodermal and endodermal neural networks of the planula larva.

Next we determined whether GLWamide neuropeptidergic input from the planula nervous system is required for metamorphic transition into a polyp in *N. vectensis*. To achieve this, we took a CRISPR-Cas9-mediated mutagenesis approach to generate animals with null alleles at the GLWamide locus. Several single guide RNA (sgRNA) species targeting the 5' and 3' regions of the gene (Materials and methods; *Figure 3*) were synthesized and microinjected with the endonuclease Cas9 into zygotes to produce double-strand DNA breaks at each of the target sites in the genome. This approach can create null mutant alleles at the target gene locus in two ways: (1) by introducing frameshift-causing indel mutations in the 5' region of the gene via an error-prone, non-homologous end-joining (NHEJ) DNA repair mechanism, so that the translated peptide sequence is altered, and (2) by deleting the protein-coding region of the gene upon cleavages at flanking sites (cf. [*Nakayama et al., 2013*]). However, injected animals–here referred to as 'F0' animals–were usually mosaic mutants (*Figure 3—figure supplement 1*; [*Servetnick et al., 2017*]); mutagenesis occurred mosaically in an F0 embryo so that individual cells would carry different mutations, or no mutation at all. Therefore, it could not be assumed that every cell in an F0 animal harbored a null mutation at the locus of interest.

We thus aimed to produce knockout mutant lines. First, F0 mosaic mutants were crossed with wild-type animals to generate F1 animals. Subsequently, F1 animals harboring a null mutant allele at the GLWamide locus were identified by genotyping each individual using genomic DNA extracted from pieces of oral tentacles. Upon screening, we found three different null mutant alleles at the GLWamide locus ($glw^{-a}$, $glw^{-b}$, and $glw^{-c}$), all of which carried frameshift mutations predicted to inhibit normal translation of neuropeptide precursor sequences (*Figure 3A–D*). glw F1 heterozygous mutants included one female with the genotype $glw^{-a}/+$ (i.e. wildtype), one female with the genotype $glw^{-b}/+$, one female with the genotype $glw^{-c}/+$, and four males with the genotype $glw^{-c}/+$. We then crossed F1 heterozygous mutants with one another to generate null mutants in, on average, a quarter of the F2 progeny. For instance, the $glw^{-b}/+$ female and the $glw^{-c}/+$ male were crossed to generate glw heterozygous null mutants with the genotype $glw^{-b}/glw^{-c}$.

To monitor development of null mutants, we first genotyped individual F2 embryos at the gastrula stage (i.e. at 1 dpf) (*Figure 3E*). Taking advantage of the ability of the oral half of the gastrula embryo to develop into a polyp (cf. [*Fritzenwanker et al., 2007*; *Lee et al., 2007*]), a piece of an aboral tissue of each F2 embryo was surgically separated and used for genomic DNA extraction,

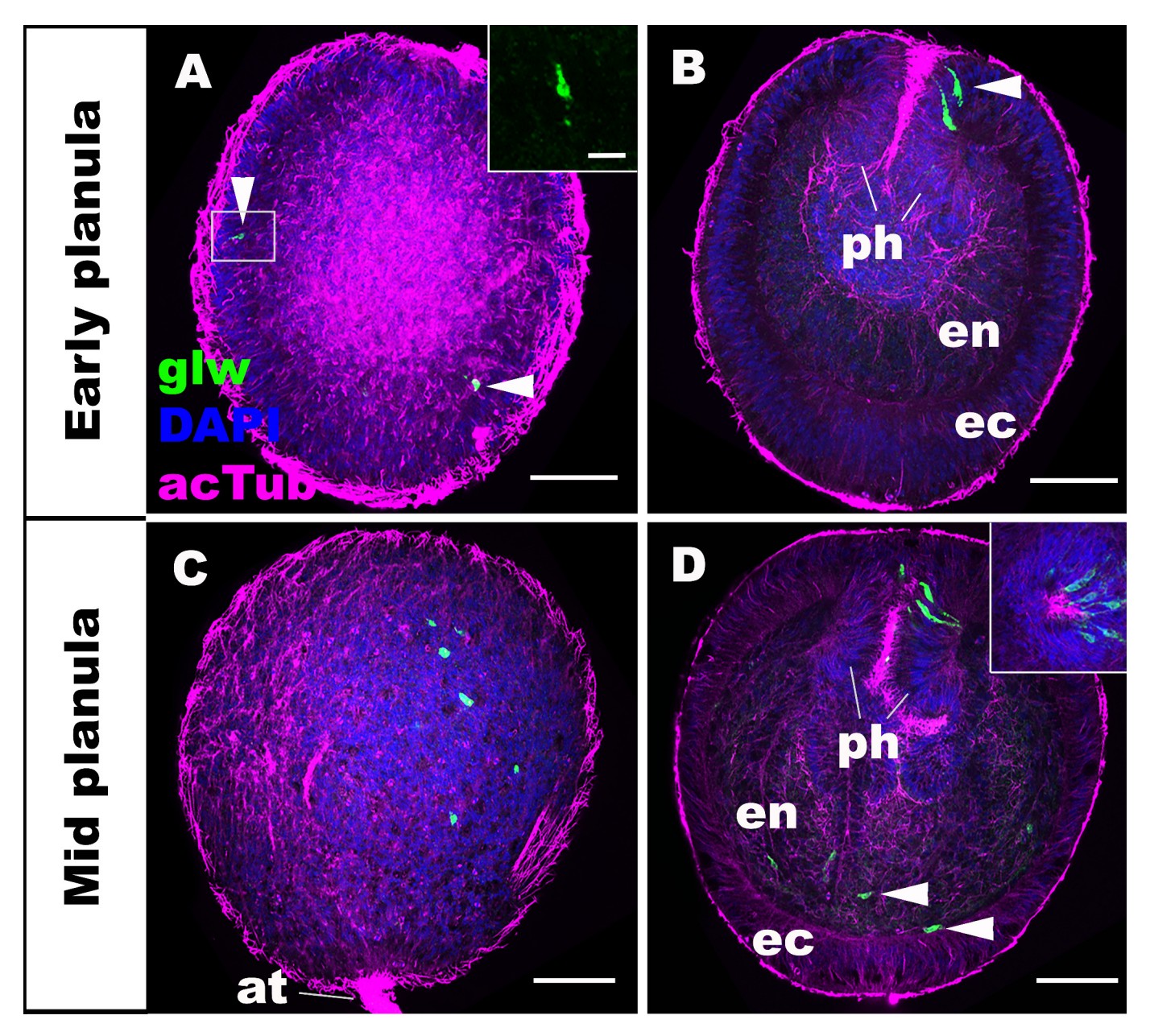

**Figure 1.** GLWamide precursor transcripts are expressed in ectodermal and endodermal epithelial cells of sea anemone planula larvae. Z-projections of confocal sections of *Nematostella vectensis* at planula stages, labeled with an antisense riboprobe against GLWamide precursor transcript ('glw') and an antibody against acetylated ∂-tubulin ('acTub'). Nuclei are labeled with DAPI. All panels show side views of animals with the blastopore/mouth facing up. The left columns (A and C) show superficial planes of section at the level of the ectoderm, and the right columns (B and D) show longitudinal sections through the center. (A, B) early planula. GLWamide transcripts are found in ectodermal sensory cells that are present in the outer epithelium and the pharynx (arrowheads in A and B). The inset in A show transcript-expressing epithelial cells in boxed regions with the apical side of the cell facing up. Note the spindle-shaped morphology characteristic of cnidarian sensory cells (*Thomas and Edwards, 1991*). (C, D) mid-planula. A subset of endodermal cells begin to express GLWamide transcript (arrowheads in D). The inset in D shows transverse sections of the pharynx, revealing the asymmetry in the spatial distribution of GLWamide-positive sensory cells. Abbreviations: ph pharynx; ec ectoderm; en endoderm; at apical tuft. Scale bar: 50 μm (A-D); 10 μm (inset in A).

DOI: https://doi.org/10.7554/eLife.39742.002

The following source data and figure supplement are available for figure 1:

**Source data 1.** A confocal image stack used to generate panels A and B (dapi_blue actub_purple glw transcript_green).

DOI: https://doi.org/10.7554/eLife.39742.004

*Figure 1 continued on next page*

Figure 1 continued

**Source data 2.** A confocal image stack used to generate panels C and D (dapi_blue actub_purple glw transcript_green).
DOI: https://doi.org/10.7554/eLife.39742.005
**Figure supplement 1.** Genomic organization, cDNA sequences, and predicted mature peptides of GLWamide gene.
DOI: https://doi.org/10.7554/eLife.39742.003

while the remaining embryo was individually maintained and allowed to regulate and develop into polyps. The genomic DNA extracted from an aboral tissue of each embryo was subsequently used as a template for PCR with allele-specific primers to determine the genotype of the embryo. The F2 embryos were then sorted by genotype, and null mutant and wildtype groups were used for subsequent analyses of development.

To determine whether GLWamide null mutants can undergo metamorphosis from a planula into a polyp in *N. vectensis*, we analyzed the morphology of the null mutants, $glw^{-b}/glw^{-c}$, and the wildtype control animals at 3dpf and 5dpf by using confocal microscopy. At 3dpf, $glw^{-b}/glw^{-c}$ null mutants appeared to be morphologically identical to normal swimming planula larvae (*Figure 4A–C,G–I*), with an aboral apical tuft (at in *Figure 4B,H*) and an endoderm in which longitudinal muscle fibers are developing (arrowheads in *Figure 4C,I*). By 5dpf, the majority of $glw^{-b}/glw^{-c}$ null mutants had transformed into morphologically normal primary polyps ($glw^{-b}/glw^{-c}$, 80% (n = 35); wildtype control, 78.6% (n = 56); *Figure 4D–F,J–L*). They show morphological hallmarks of metamorphosis, including the development of parietal and retractor muscles in the endoderm (pm and rm in *Figure 4D,J*), the loss of the apical tuft (arrowheads in *Figure 4E,K*; [*Nakanishi et al., 2012*]), and the formation of oral tentacles with polyp-specific cell types such as the hair cells (*Figure 4F,L*; (*Nakanishi et al., 2012*; *Watson et al., 2009*). Other *glw* null mutants–namely, $glw^{-a}/glw^{-c}$ (F2) and $glw^{-a}/glw^{-a}$ (F3)— also metamorphosed (data not shown). These results demonstrate that GLWamide of the larval nervous system is dispensable for metamorphosis in *N. vectensis,* and that cnidarian metamorphosis can occur via a GLWamide-independent mechanism.

Because it has been proposed that the original function of (neuro)peptides, before emergence of the nervous system, was to regulate regenerative responses upon injury and/or stress as growth factors (*Moroz, 2009*), we also examined the regenerative capacity of GLWamide knockout mutants. F2 $glw^{-b}/glw^{-c}$ and wildtype control juvenile polyps were transversely cut into oral and aboral halves, which were given two weeks to regenerate the aboral and oral regions, respectively. Regeneration occurred normally in $glw^{-b}/glw^{-c}$ mutants; the oral half regenerated the aboral region ($glw^{-b}/glw^{-c}$, 100% (n = 6); wildtype control, 83.3% (n = 6), and the aboral half regenerated the oral structures such as the pharynx and oral tentacles ($glw^{-b}/glw^{-c}$, 100% (n = 6); wildtype control, 100% (n = 6). We also note that although pharmacological evidence suggests a role in oocyte maturation and spawning in a hydrozoan cnidarian (*Takeda et al., 2013*), GLWamide knockout mutants can generate functional gametes in response to light and temperature cues provided to induce spawning. Thus, GLWamides are not essential for regeneration or reproduction in *N. vectensis.*

We next considered the possibility that GLWamide might act to accelerate metamorphosis. We tested this alternative hypothesis by examining the time course of polyp formation in GLWamide null mutants relative to that in wildtype control animals. For this experiment, we crossed F2 $glw^{-a}/glw^{-c}$ adults with each other to generate F3 null mutants, and F2 $glw^{+}/glw^{+}$ adults with each other to generate F3 wildtype control animals (*Figure 5—figure supplement 1*); $glw^{-a}/glw^{-c}$ and $glw^{+}/glw^{+}$ F2 adults were siblings. Possible genotypes of null mutants examined were $glw^{-a}/glw^{-a}$, $glw^{-a}/glw^{-c}$, or $glw^{-c}/glw^{-c}$. Eggs of null mutants and wildtype control animals were fertilized by dropping egg packages into sperm-containing water at the same time (i.e. within a one-minute time window). Fertilized eggs were subsequently de-jellied, and animals were raised in constant darkness at 16°C; development at this temperature occurs slower than that at room temperature (20–25°C). At 2 dpf, healthy-appearing animals (i.e. without tissue damage) were transferred into wells of 24 well plates, each containing approximately 20 individuals in 1 ml of 1/3 seawater (wildtype, n = 6 experiments; mutants, n = 9 experiments). Half of the water in each well was replaced daily. For each experiment, we quantified the proportions of animals that were at the tentacle-bud or primary polyp stage from 6 dpf to 10 dpf. We observed no detectable difference in larval morphology at 5 dpf between null mutants and wildtype control animals, and thus we assume that larval development occurs normally at the morphological level. We used the number of polyps counted at 17 dpf as the total number of

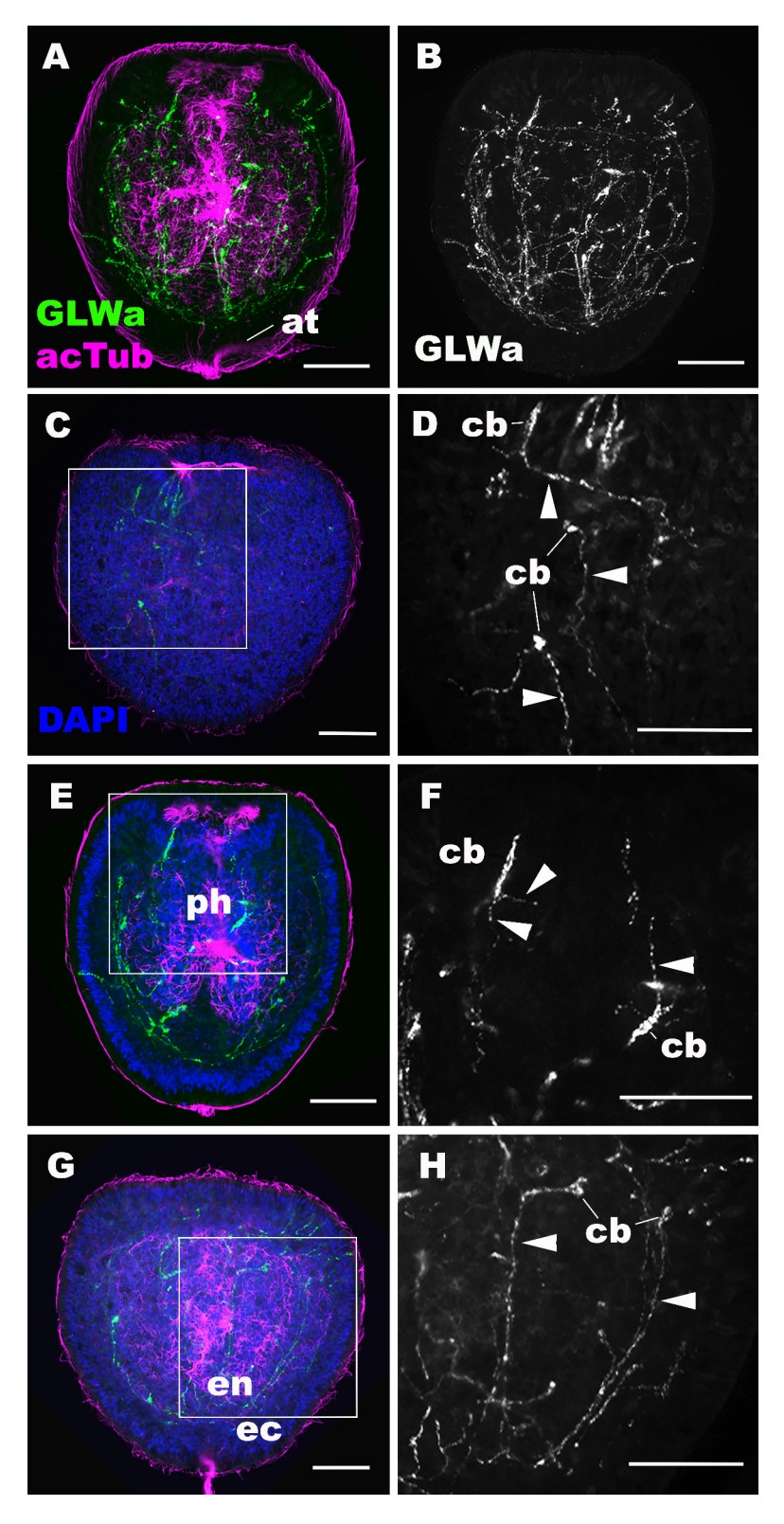

**Figure 2.** GLWamide-positive epithelial neurons form ectodermal and endodermal nervous systems of sea anemone planula larvae. Z-projections of confocal sections of *Nematostella vectensis* at mid-late planula stages, labeled with antibodies against GLWamide ('GLWa') and acetylated ∂-tubulin ('acTub'). Nuclei are labeled with DAPI. All panels show side views of animals with the blastopore/mouth facing up. (A and B) show sections through the entire animals. (C and D) show superficial planes of section at the level of the surface ectoderm. (E and F) show longitudinal sections through the center

*Figure 2 continued on next page*

*Figure 2 continued*

at the level of the pharyngeal sensory cells. (**G and H**) show longitudinal sections at the level of the endodermal neurons. Boxed regions in C, E and G are magnified in D, F and H, respectively. Note that ectodermal neurons in the surface ectoderm extend neurites laterally and aborally, but not orally (arrowheads in D), while pharyngeal ectodermal neurons extend neurites in all three directions (arrowheads in F). Endodermal neurons extend neurites that are part of longitudinally oriented neuronal bundles (arrowheads in H). Abbreviations: at apical tuft; cb cell body; ph pharynx; ec ectoderm; en endoderm. Scale bar: 50 μm.

DOI: https://doi.org/10.7554/eLife.39742.006

The following source data and figure supplement are available for figure 2:

**Source data 1.** A confocal image stack used to generate panels A, B, E and F (dapi_cyan actub_green GLWa_red).

DOI: https://doi.org/10.7554/eLife.39742.008

**Source data 2.** A confocal image stack used to generate panels C, D, G and H (dapi_cyan actub_green GLWa_red).

DOI: https://doi.org/10.7554/eLife.39742.009

**Figure supplement 1.** The anti-GLWamide antibody used in this study labels a subpopulation of ectodermal sensory cells of the rhopalial nervous system in the scyphozoan cnidarian *Aurelia sp.1*.

DOI: https://doi.org/10.7554/eLife.39742.007

animals competent to become polyps for each experiment; the remaining animals that have not undergone metamorphosis by 17 dpf showed signs of abnormal development, and thus were excluded from the analysis. On average 17.5 animals, ranging from 14 to 20, were examined per experiment. We found that the wildtype control animals began to metamorphose at 6dpf, and the majority (~86%) had begun transforming by 8 dpf, while the null mutants began to metamorphose at 7dpf, and less than half (~42%) had begun metamorphosis by 8 dpf (*Figure 5*); the differences at 7 and 8 dpf were statistically significant ($\alpha = 0.05$). Thus GLWamide null mutants metamorphosed at a slower rate than the wildtype control.

To confirm that GLWamide deficiency is indeed responsible for this developmental delay, we tested whether treatment of GLWamide null mutants with synthetic GLWamide peptides (CEA-GAP<u>GLWamide</u>), which were used to generate the anti-GLWamide antibody, would be sufficient to rescue the phenotype in developmental timing. Incubation of the mutants (n = 9 experiments) with 10 μM GLWamide began at 2 dpf and continued through 10 dpf with half of the peptide-containing media being replaced daily. On average 17.2 animals, ranging from 12 to 23, were examined per experiment. We observed that, similar to the wildtype animals, the null mutants treated with GLWamide began metamorphosis at 6 dpf, and the majority (~84%) had started metamorphosis by 8 dpf, showing statistically higher proportions of metamorphosing animals at 7 and 8 dpf relative to untreated null mutants ($\alpha = 0.05$; *Figure 5*). Taken together, we conclude that GLWamide regulates metamorphosis by accelerating the process in *N. vectensis*.

## Discussion

Here we created the first neuropeptide knockout animals from an early-evolving animal group Cnidaria, and examined the endogenous function of deeply conserved (GL)Wamide neuropeptides in the sea anemone cnidarian *Nematostella vectensis*. We report that in *N. vectensis* GLWamide neuropeptides are expressed in ectodermal and endodermal nervous systems of the planula larva, but are not essential for transformation of the planula into a polyp. Instead we find that GLWamides regulate the timing of polyp formation by accelerating the metamorphic process in *N. vectensis*. These results raise the possibility that the ancestral role of Wamide neuropeptides in animals may have been to regulate the timing of life cycle transition.

Indeed, several lines of evidence indicate that the Wamide neuropeptide is an accelerator, but not an essential regulator, of metamorphosis in animals. First, as described above, exogenous Wamides can trigger life cycle transition across annelids and cnidarians (*Conzelmann et al., 2013*; *Erwin and Szmant, 2010*; *Iwao et al., 2002*; *Leitz et al., 1994*; *Schmich et al., 1998b*), and our results show slower rates of life cycle transition in GLWamide mutant larvae in sea anemones. These data support Wamide's role as an accelerator of metamorphosis across animals.

Second, Wamides do not appear necessary for life cycle transition in animals where functional perturbation data are available. For instance, although in the annelid polychaete *Platynereis,* MIP (Wamide) receptor expression is necessary for MIP-induced larval settlement behavior

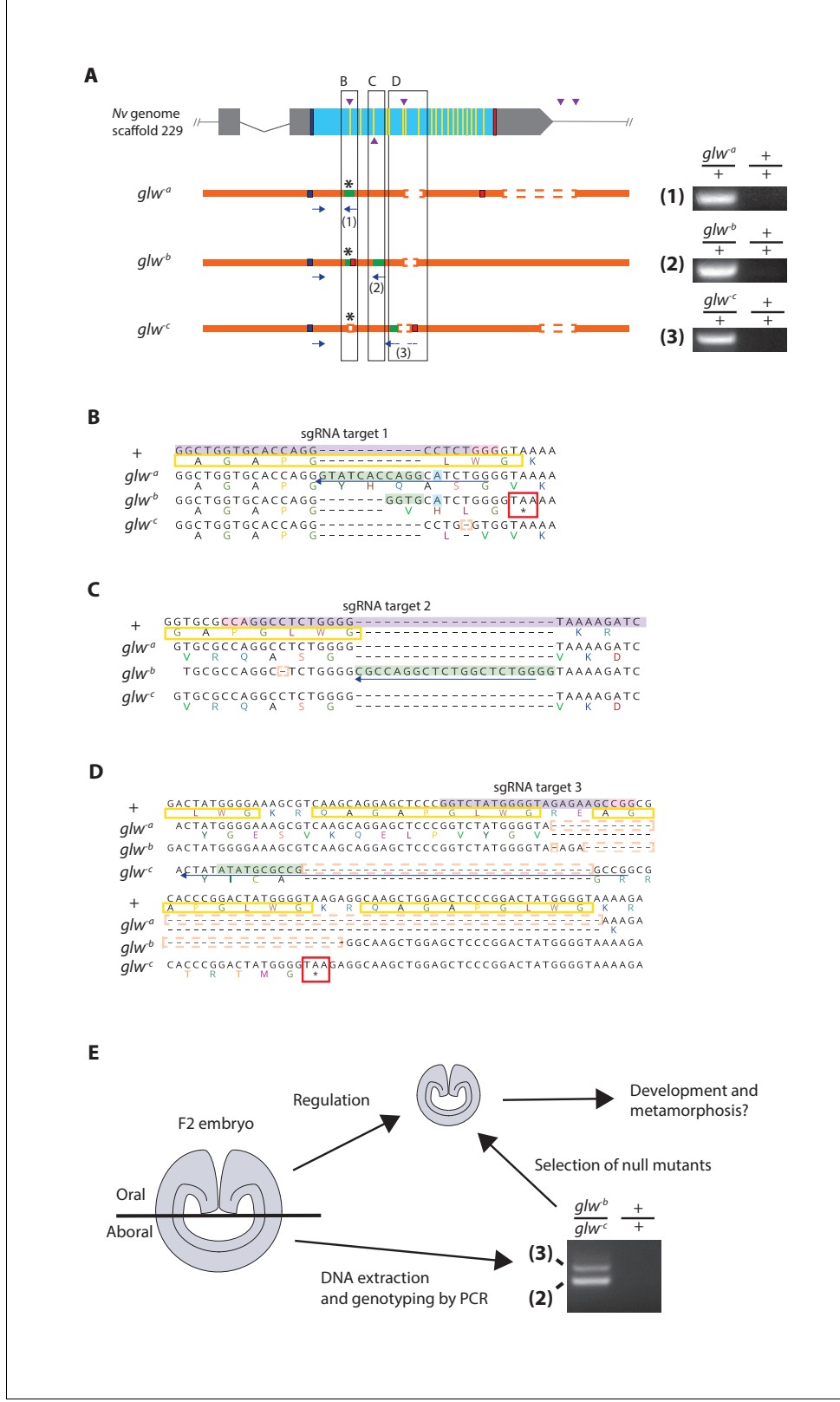

**Figure 3.** Generation of GLWamide null mutants by CRISPR-Cas9-mediated mutagenesis and F1 heterozygous mutant crosses. (**A**) Schematic views of the GLWamide locus and mutant alleles (*glw⁻ᵃ*, *glw⁻ᵇ* and *glw⁻ᶜ*). Blue bars show predicted translation start sites; red bars show predicted translation termination sites; yellow bars show predicted GLWamide-encoding regions (*Figure 1—figure supplement 1*). Purple arrowheads show sgRNA target sites. Insertions are labeled in green, and deletions are shown blank. Asterisks show the sites of frameshift-causing mutations. Blue arrows mark regions

*Figure 3 continued on next page*

*Figure 3 continued*

targeted in the PCR analysis shown to the right; reverse primers are numbered (1 - 3). Note that mutant allele-specific primers (1 - 3) generate PCR products from genomic DNA samples of heterozygous mutants, but not from those of wildtype animals. (B-D) Nucleotide and translated amino acid sequences of wild-type and mutant alleles boxed in A. sgRNA target sites are shown in purple, and PAM sites are shown in pink. Predicted translation start sites are boxed in blue; predicted translation termination sites are boxed in red. Insertions are labeled in green, replacements in blue, and deletions boxed in dotted orange lines. Predicted immature neuropeptide sequences are boxed in yellow (cf. *Figure 1—figure supplement 1*). (E) a schematic showing the procedure of genotyping F2 embryos for developmental analyses. F2 embryos generated by F1 heterozygous mutant crosses were transversely bisected along the oral-aboral axis, and the oral halves were allowed to regulate and develop into polyps. Genomic DNA was extracted from single aboral halves and was used to genotype each embryo by PCR. The primers (2) and (3) were used to identify *glw⁻ᵇ/glw⁻ᶜ* embryos. '+' indicates a wildtype allele.

DOI: https://doi.org/10.7554/eLife.39742.010

The following figure supplement is available for figure 3:

**Figure supplement 1.** F0 embryos are mosaic mutants.

DOI: https://doi.org/10.7554/eLife.39742.011

(*Conzelmann et al., 2013*), reducing the expression of MIP by morpholino antisense oligonucleotide inhibits feeding behavior in the juvenile, but not metamorphosis (*Williams et al., 2015*). This could mean that there is a compensatory mechanism, possibly through as-yet-unknown ligands for the 'MIP' receptor. Alternative possibilities that silencing of the MIP gene was incomplete and the residual MIPs were sufficient for triggering the transition, or that MIP does not endogenously regulate larval settlement and metamorphosis, cannot be ruled out, however; application of knockout approaches as employed in this study will be necessary to test these hypotheses. It is also possible that metamorphosis is decoupled from larval settlement so that MIP-regulated settlement behavior is not essential for metamorphosis in *Platynereis*. In the hydrozoan cnidarian *Hydractinia echinata* it is noteworthy that in one of the three experimental conditions where RNAi was used to knockdown

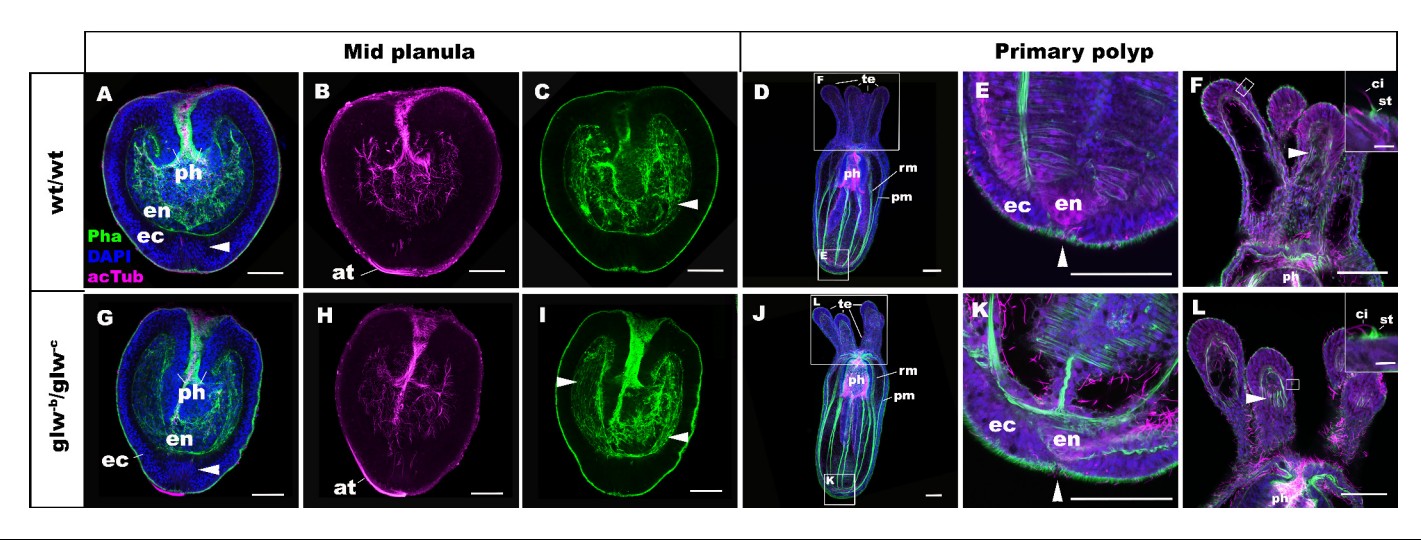

**Figure 4.** GLWamide null mutant planulae transform into morphologically normal primary polyps. Confocal sections of wild-type (**A-F**) and mutant (*glw⁻ᵇ/glw⁻ᶜ*, **G-L**) *Nematostella vectensis*, labeled with an antibody against acetylated ∂-tubulin (acTub). Filamentous actin is labeled with phalloidin (Pha), and nuclei are labeled with DAPI. All panels show side views of animals with the blastopore/mouth facing up. (**A-C, G-I**): mid-planula. Similar to the wild-type, *glw⁻ᵇ/glw⁻ᶜ* mutant planulae develop apical tufts (at) at the aboral pole; in addition, the basal translocation of nuclei in the aboral-most ectoderm (arrowheads in A, G, M), as well as the development of myofilaments in the endoderm (arrowheads in C, I), are evident. (**D-F, J-L**) primary polyp. *glw⁻ᵇ/glw⁻ᶜ* mutants develop into morphologically normal primary polyps, forming a set of four oral tentacles (**D, J**) with longitudinally oriented muscle fibers in the endoderm (arrowheads in F, L) and hair cells in the ectoderm–a polyp-specific cell type in *N. vectensis* (*Nakanishi et al., 2012*; *Watson et al., 2009*)– characterized by a single cilium (ci) surrounded at the base by stereocilia (st) (insets in F, L). Note also the development of eight sets of longitudinally oriented parietal (pm) and retractor (rm) muscles in the endoderm of the body column (**D, J**), and the loss of apical tufts (arrowheads in E, K), in wild-type as well as *glw⁻ᵇ/glw⁻ᶜ* mutant animals. Abbreviations: ph pharynx; ec ectoderm; en endoderm. Scale bar: 50 μm; 5 μm (inset in F, L).

DOI: https://doi.org/10.7554/eLife.39742.012

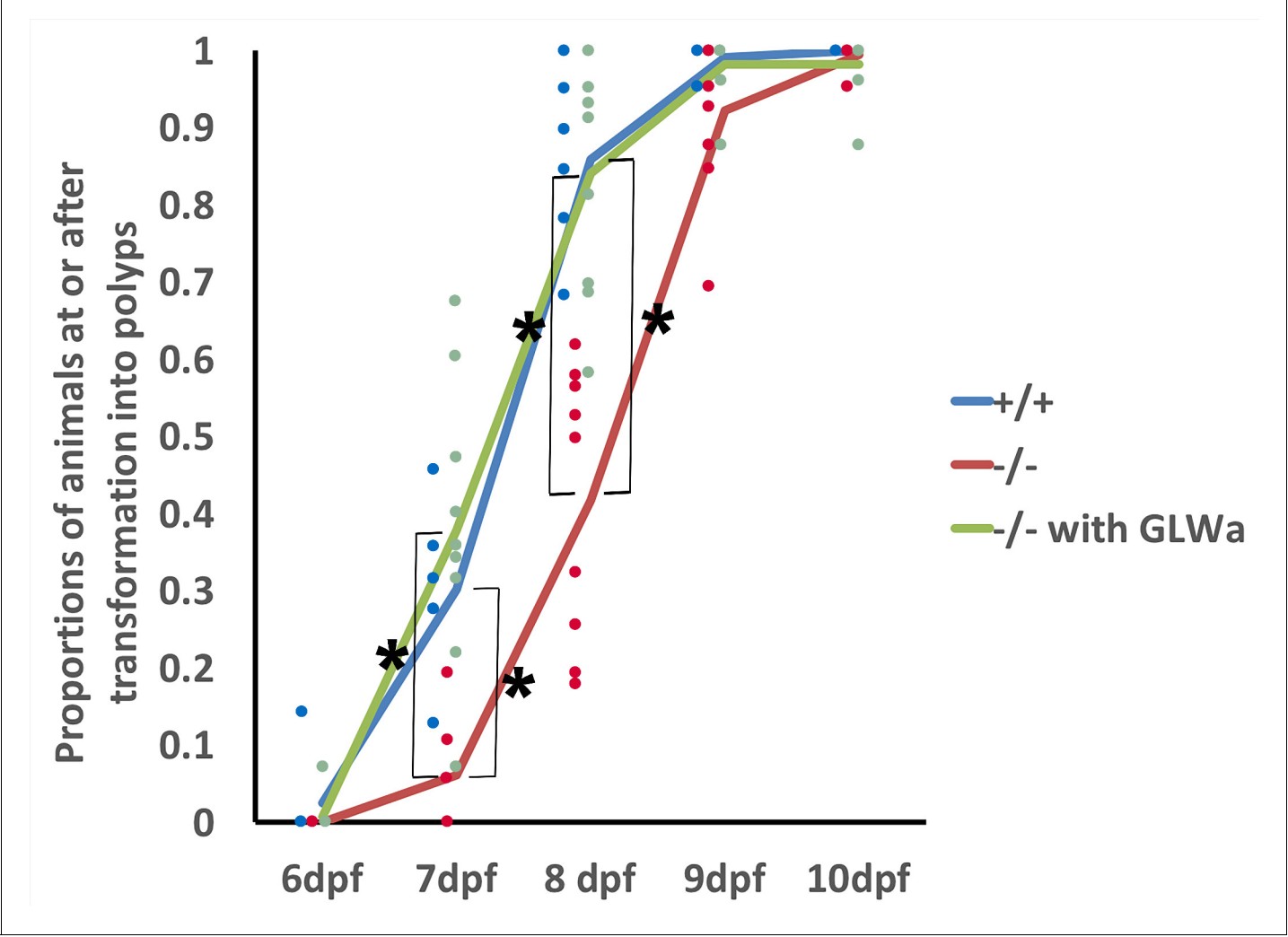

**Figure 5.** GLWamides regulate the timing of metamorphosis in sea anemone planula larvae. A line graph showing changes in the proportions of animals at the tentacle-bud or primary polyp stage from 6 dpf to 10 dpf in F3 GLWamide null mutants ('-/-'), F3 wildtype control animals ('+/+'), and F3 GLWamide null mutants treated with synthetic GLWamide peptides ('-/- with GLWa') at 16°C. The proportions of metamorphosing and metamorphosed animals were significantly lower in null mutants at 7 and 8 dpf relative to the wildtype controls and null mutants treated with GLWamide (one-tailed t-test: -/- relative to +/+, 7 dpf $p<10^{-3}$, 8 dpf $p<10^{-4}$; -/- relative to -/- with GLWa, 7 dpf $p<10^{-3}$, 8 dpf, $p<10^{-4}$). Total numbers of experiments: -/- n = 9; +/+ n = 6; -/- with GLWa n = 9. Filled circles represent data points (+/+ in blue, -/- in red and -/- with GLWa in green). * denotes statistically significant difference ($\alpha$ = 0.05).
DOI: https://doi.org/10.7554/eLife.39742.013

The following source data and figure supplement are available for figure 5:

**Source data 1.** Numerical data represented in *Figure 5*.
DOI: https://doi.org/10.7554/eLife.39742.015

**Figure supplement 1.** F3 GLWamide null mutant planulae transform into polyps.
DOI: https://doi.org/10.7554/eLife.39742.014

GLWamide expression, the rate of metamorphosis was significantly reduced at 24 hr after induction (by CsCl), but it increased to a normal level 24 hr later (*Plickert et al., 2003*). This observation is consistent with a role of GLWamide as an accelerator, but not an essential regulator, of metamorphosis; however, an alternative possiblity that GLWamide expression recovered at later stages (due to reagent degradation and/or dilution) cannot be ruled out without gene knockout data. In the present study, our gene knockout data clearly demonstrate that GLWamides are not required for life

cycle transition in the sea anemone cnidarian *N. vectensis*. We note that our data leave open the possibility that other amidated peptides could be necessary for metamorphosis in *N. vectensis*.

Third, animals that undergo metamorphosis do not necessarily have Wamide neuropeptides. For example, deuterostome bilaterians lack Wamide homologs (*Conzelmann et al., 2013*) despite metamorphosis being common in some of the lineages (e.g. echinoderms, hemichordates, ascidians, and vertebrates). Similarly, members of Porifera (sponges)—an outgroup to Eumetazoa—typically undergo metamorphosis, and yet Wamides have not been found in a sponge genome (*Srivastava et al., 2010*). Thus Wamide-independent mechanisms of metamorphosis must exist in some animal lineages.

Taken together, comparative evidence suggests that the ancestral Wamide neuropeptidergic signaling was not an indispensable component of the developmental regulatory system, but may have functioned to fine-tune the timing of life cycle transition by accelerating the developmental process—possibly in response to specific environmental cues—in the last common ancestor of Eumetazoa. Understanding how ancestral Wamides may have controlled developmental timing—the nature of the sensory stimuli that trigger a release of Wamides, and identities of Wamide receptors and target developmental regulatory genes, in particular—requires further investigation into Wamide function across Eumetazoa.

In addition to developmental timing, Wamides appear to be involved in regulating organismal behavior in extant animals. For instance, treatment with MIPs can suppress spontaneous contractions of the hindgut in insects (*Schoofs et al., 1991*). In the annelid *Platynereis,* in contrast, MIPs increase gut peristalsis and are necessary for feeding behavior (*Williams et al., 2015*). In cnidarians, exogenous GLWamides can induce muscle contractions in hydrozoan and anthozoan polyps (*Takahashi et al., 2003*) and stimulate migration of planula larvae in *Hydractinia* (*Katsukura et al., 2004*); however, it remains to be confirmed whether GLWamides endogenously control these behavioral processes. Demystifying the endogenous behavioral function(s) of deeply conserved neuropeptides—Wamides as well as others (e.g. RFamides)—across Eumetazoa will be important for illuminating how the primitive nervous system of the last common ancestor of Eumetazoa may have coordinated organismal behavior—another fundamental problem of neural evolution that remains enigmatic.

## Materials and methods

**Key resources table**

| Reagent type (species) or resource | Designation | Source or reference | Identifiers | Additional information |
|---|---|---|---|---|
| Biological sample (*Nematostella vectensis*) | F2 glw-a/glw-c | this paper | Nv F2 glw-a/glw-c_ this paper | See the Results section of the paper for a description of the biological sample. |
| Antibody | Anti-GLWamide (rabbit) | this paper | Anti-GLWamide_this paper:AB_2737386 | (1:200) See the Materials and Methods section of the paper for a description of the antibody. |

### Animal culture

*Nematostella vectensis* were cultured as previously described (*Fritzenwanker and Technau, 2002*; *Hand and Uhlinger, 1992*).

### RNA extraction, cDNA synthesis and gene cloning

Total RNA was extracted from a mixture of planulae and primary polyps using TRIzol (Thermo Fisher Scientific). cDNAs were synthesized using the BD SMART RACE cDNA Amplification Kit (BD Biosciences, San Jose, CA, USA). GLWamide gene sequence (*Anctil, 2009*) was retrieved from the Joint Genome Institute genome database (Nematostella vectensis v1.0; http://genome.jgi-psf.org/Nemve1/Nemve1.home.html; gene ID number 126270). 5' and 3' RACE were conducted in order to confirm in silico predicted gene sequences, and to generate templates for RNA probes for in situ hybridization experiments. RACE PCR fragments were cloned into the pGEM-T plasmid vector using the pGEM-T Vector Systems (Promega), and were sequenced at Macrogen Corp., Maryland.

## Generation of an antibody against GLWamide

An antibody against a synthetic peptide CEAGAPGLWamide corresponding in amino acid sequence to GLWamide I (*Figure 1—figure supplement 1*) was generated in rabbit (YenZym Antibodies, LLC). Following immunization, the resultant antiserum was affinity purified with the CEAGAPGLWamide peptide. The affinity purified antibody was then affinity-absorbed with KECPPGLWGC-cooh and KECLPGVWG-cooh, which correspond to predicted peptides generated by NvLWamide-like (Nv242283), to produce a GLWamide-specific antibody. However, this antibody likely cross-reacts with GLWamides regardless of the N-terminal sequence, as immunohistochemistry in *Aurelia* ephyrae (larval medusae) shows a staining pattern consistent with that previously documented by using an antibody against EQPGLWamide (*Figure 2—figure supplement 1*; [*Nakanishi et al., 2009*]).

## CRISPR-Cas9-mediated mutagenesis

Using the published method (*Nakayama et al., 2013*), sgRNAs were in vitro transcribed from PCR-amplified templates by using MEGAscript transcription kits (Ambion). To generate sgRNA templates, we designed 5' and 3' oligonucleotides as follows.

5' oligonucleotide:

5'-AAT**TAATACGACTCACTATA***Gn$_{18-19}$*GTTTTAGAGCTAGAAATAGC-3' where the RNA promoter (T7 is shown) is in bold and the sgRNA target sequence is in *italic*. The first G in the sgRNA target sequence is necessary for transcription by RNA polymerase.

3' oligonucleotide:

5'-GATCCGCACCGACTCGGTGCCACTTTTTCAAGTTGATAACGGACTAGCCTTATTTTAACTTGCTATTTCTAGCTCTAAAAC-3'

19–20 nt-long sgRNA target sites were identified in genomic locus for NvGLW by using the Targeter website (http://zifit.partners.org) (*Hwang et al., 2013*). To minimize off-target effects, target sites that had 17 bp-or-higher sequence identity elsewhere in the genome (*Nematostella vectensis v1.0*; http://genome.jgi.doe.gov/Nemve1/Nemve1.home.html) were excluded. Selected target sequences were as follows.

5'-GGCTGGTGCACCAGGCCTCT-3' (Cr1)
5'-GATCTTTTACCCCAGAGGCC-3' (Cr2)
5'-GGTCTATGGGGTAGAGAAGC-3' (Cr3)
5'-GGGCTGCGTTATACTTGTCT-3' (Cr4)
5'-GGTACACTCTAACAGATTGT-3' (Cr5)

sgRNA species were mixed at equal concentrations, and the sgRNA mix and Cas9 endonuclease (PNA Bio, PC15111, Thousand Oaks, CA, USA) were co-injected into fertilized eggs at concentrations of 500 ng/μl and 1000 ng/μl, respectively.

## Genotyping of embryos

Genomic DNA from single embryos or pieces of polyp tentacles from single polyps was extracted by using a published protocol (*Ikmi et al., 2014*), and the targeted genomic loci were amplified by PCR. To determine the sequence of mutant alleles, PCR products from genomic DNA extracted from F1 mutant polyps were gel-purified, cloned and sequenced by using a standard procedure. Using the sequence information of mutant alleles, allele-specific primers were designed in order to genotype single F2 embryos. Primers used are listed below.

For sequencing mutant alleles,
Forward 5'- AATGTGAACAACGACGACGACAACG-3'
Reverse 5'- GGAGTGGTTTCCAAATCTCCCGAGC −3'
For genotyping,
Forward 5'- CATGCGGAGACCAAGCGCAAGGC-3' (*Figure 3A*)
Reverse (*glw$^{-a}$*) 5'-CCAGATGCCTGGTGATAC-3' ([1] in *Figure 3A*)
Reverse (*glw$^{-b}$*) 5'- CCCAGAGCCAGAGCCTGGCG-3' ([2] in *Figure 3A*)
Reverse (*glw$^{-c}$*) 5'- CGGCCGGCGCATATATAG-3' ([3] in *Figure 3A*)

## Immunofluorescence, in situ hybridization, and confocal microscopy

Animal fixation, immunohistochemistry, and in situ hybridization were performed as previously described (*Nematostella vectensis*, (**Martindale et al., 2004**; **Nakanishi et al., 2012**); *Aurelia sp.1* (**Nakanishi et al., 2009**)). For immunohistochemistry, we used primary antibodies against GLWamide (rabbit, 1:200), acetylated ∂-tubulin (mouse, 1:500, Sigma T6793) and tyrosinated ∂-tubulin (mouse, 1:500, Sigma T9028), and secondary antibodies AlexaFluor 568 (mouse. 1:200, Molecular Probes) and AlexaFluor 647 (mouse, 1:200, Molecular Probes). Nuclei were labeled using fluorescent dyes DAPI (1:1,000, Molecular Probes), and filamentous actin was labeled using AlexaFluor 488-conjugated phalloidin (1:25, Molecular Probes). For in situ hybridization, antisense digoxigenin-labeled riboprobes were synthesized by using the MEGAscript transcription kits according to manufacturer's recommendation (Ambion), and were used at the final concentration of 1 ng/µl. Fluorescent images were recorded using a Zeiss LSM 710 or Leica SP5 Confocal Microscopes. Images were viewed using ImageJ.

## Acknowledgements

We are grateful to David Simmons for his help with the experimental design of CRISPR-Cas9 assay, and to other members of the Martindale laboratory for constructive discussion. We would also like to thank Sai Divya Challapalli of the Nakanishi laboratory for animal husbandry, and anonymous reviewers for comments on the earlier version of the manuscript, which greatly improved the manuscript. This work was supported by the NASA Astrobiology Postdoctoral Fellowship (to NN), funds from the University of Arkansas (to NN), and a NASA Astrobiology Institute Grant (to MQM).

## Additional information

### Funding

| Funder | Author |
| --- | --- |
| National Aeronautics and Space Administration | Nagayasu Nakanishi Mark Q Martindale |
| University of Arkansas | Nagayasu Nakanishi |

The funders had no role in study design, data collection and interpretation, or the decision to submit the work for publication.

### Author contributions

Nagayasu Nakanishi, Conceptualization, Resources, Data curation, Formal analysis, Funding acquisition, Validation, Investigation, Methodology, Writing—original draft, Project administration, Writing—review and editing; Mark Q Martindale, Conceptualization, Resources, Supervision, Funding acquisition, Methodology, Writing—review and editing

### Author ORCIDs

Nagayasu Nakanishi (iD) http://orcid.org/0000-0001-7516-5078
Mark Q Martindale (iD) https://orcid.org/0000-0002-5805-5640

### Decision letter and Author response

Decision letter https://doi.org/10.7554/eLife.39742.020
Author response https://doi.org/10.7554/eLife.39742.021

## Additional files

### Supplementary files

• Transparent reporting form
DOI: https://doi.org/10.7554/eLife.39742.016

## Data availability

All data generated or analyzed during this study are included in the manuscript and supporting files. A source data file has been provided for Figures 1, 2 and 5. The GLWamide gene sequence has been deposited to GenBank under accession number MH939200.

The following dataset was generated:

| Author(s) | Year | Dataset title | Dataset URL | Database and Identifier |
|---|---|---|---|---|
| Nagayasu Nakanishi, Mark Q Martindale | 2018 | Nematostella vectensis GLWamide precursor, mRNA, complete cds | https://www.ncbi.nlm.nih.gov/nuccore/1483577241 | GenBank, MH939200 |

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
