## [Decision Letter]

Thank you for submitting your article "CRISPR knockouts reveal an endogenous role for ancient neuropeptides in regulating developmental timing in a sea anemone" for consideration by *eLife*. Your article has been reviewed by Diethard Tautz as the Senior Editor, Alejandro Sánchez Alvarado as the Reviewing Editor, and three reviewers. The reviewers have opted to remain anonymous.

The reviewers have discussed the reviews with one another and the Reviewing Editor has drafted this decision to help you prepare a revised submission.

Summary:

Nakanishi and Martindale use a knockout approach to study the role of GLWamide neuropeptides play in polyp morphogenesis in the Cnidarian, *Nematostella vectensis*. The authors find that GLWamides are not required for the polyp transition but significantly accelerate polyp morphogenesis. Because of the ancient origin of these peptides, the authors propose that Wamide peptides likely have an evolutionarily-conserved role in accelerating developmental transitions in both cnidarians and bilaterians. While many of the findings are not unexpected based on past studies and could, therefore, be viewed as incremental, we are enthusiastic and view the study as innovative and highly important because it: (1) uses modern, definitive genetic techniques to address this question for the first time; and (2) provides a roadmap for using *Nematostella* for complex reverse genetic experiments. In fact, this is one of the first studies that has used F2 and F3 non-mosaic CRISPR KO animals (rather than mosaic F0) in a cnidarian. As such, this work sets a new standard for cnidarian loss-of-function research. Progress in our understanding of the evolution of physiological systems has been hampered by lack of tractable cnidarian model organisms, so it is exciting to see this kind of study.

Essential revisions:

Because of the importance of this study, we would like for the authors to address the following issues:

1) The results that were obtained from the experiments reported are interesting; however, the authors' conclusion is based on the assumption that the gene in question is the only GLWamide-encoding gene in *Nematostella*. What is the evidence that no other related peptide is present (i.e., very short seq: LWG followed by K or R; easy to overlook), given also that the current genome assembly isn't great (see Figure 5—figure supplement 1)?

2) The antibody they used was generated against CEAGAPGLWamide and would probably not detect other Wamide-like peptides. The delayed metamorphosis reported by the authors is also consistent with the presence of an additional XWamide/LWamide that may induce metamorphosis less effectively. Also, *Nematostella* metamorphosis is different from other studied cnidarians (hydrozoans and corals) in that it does not require an exogenous trigger. Hence, these findings could be true for *Nematostella* but not necessarily for other cnidarians, where a microbial film or LWamide change metamorphosis rates from <5% in their absence to >95% when applied. Our recommendation: either provide evidence to exclude other LWamide encoding genes in *Nematostella* or, adapt the conclusion to the actual findings.

3) The paper would benefit from a better discussion of previous work done on the role of LWamides in cnidarians. For example, McFarlane et al. (1987), Takahashi et al. (1997), and Katsukura et al. (2004) reported that LWamide peptides (also known in Hydra as hym-54) cause a rhythmic contraction in sea anemone and hydrozoan muscles (at lower concentrations). Has there been any defect in locomotion or phototaxis in KO animals?

4) The authors write that previous work suggested the Wamides are required for metamorphosis. This is not entirely correct, and the text should be carefully revised. Conzelmann et al. showed that exogenous MIP can 'induce' or 'trigger' larval settlement and that the effects on the cilia require the MIP-receptor. They did not claim that MIP is 'required' for settlement.

5) In the Discussion section, the authors refer to a study by Plickert et al. on the role of GLWamide peptides in regulating *Hydractinia* metamorphosis. Plickert et al. concluded that GLWamide was required for metamorphosis in *Hydractinia*. This was based on two lines of evidence. One was the RNAi knockdown of the proneuropeptide that led to a delay/reduced rate in metamorphosis. The other was the pharmacological inhibition of the α-amidating enzyme that is required to convert the C-terminal Gly into an amide group during neuropeptide maturation. Inhibition of this enzyme with the copper chelators diethyl-dithiocarbamate (DDC) or tetra-ethyl-thiuram-disulfide (TETD) completely blocked metamorphosis. This effect could be rescued by synthetic GLWamide. Nakanishi and Martindale only mention the RNAi results and argue that the data in the Plickert et al. paper are consistent with GLWamide having an accelerating role. However, they should also discuss the inhibitor results and the rescue by the peptide, which suggest a requirement of amidated peptides.

6) It may well be that there are species-specific differences in the involvement of GLWamides in the regulation of metamorphosis. The discussion presents the role of GLWamides and MIPs in general, assuming a conserved role. It should be made clear when the authors talk about the function of the peptide in *Nematostella* and when they suggest a conservation of function in other species.

7) The following two statements should be carefully reworded:

Introduction: the authors write that "Contrary to the hypothesis that Wamide is necessary for life cycle transition, we report that Wamide knockout mutant larvae can transform into morphologically normal polyps."

and

Results section: "The above results are incompatible with the previously proposed hypothesis that GLWamide is an indispensable component of the regulatory system controlling cnidarian metamorphosis (e.g. Plickert et al., 2003)."

This work was done in *Nematostella* and the hypothesis that Wamide is necessary for lifecycle transition was proposed based on experiments in *Hydractinia*. The two diverged in or prior to the Cambrian. See also previous comments about the copper chelator experiments in *Hydractinia*.

8) The authors write that "in the annelid polychaete *Platynereis*, MIP (Wamide) receptor expression is necessary for larval settlement behavior (Conzelmann et al., 2013) … reducing the expression of MIP by morpholino antisense oligonucleotide inhibits feeding behavior in the juvenile, but not metamorphosis (Williams, Conzelmann, and Jekely, 2015). This suggests the presence of as-yet-unknown ligands for the "MIP" receptor." The conclusion that the *Platynereis* MIP receptor has other ligands is not supported by the available data. First, the morpholino-mediated knockdown of the MIP receptor by Conzelmann et al. only abrogated the effect of MIP on ciliary closures. A longer-term effect on spontaneous settlement behavior (i.e. without extra MIP addition) was not investigated. So, the statement "receptor expression is necessary for larval settlement" is not precise. Likewise, the observation that MIP knockdown by morpholinos inhibits feeding behavior in the juvenile, but not metamorphosis, does not mean that the MIP receptor has other ligands. Instead, what has been shown recently, is that MIP peptides signal through two different receptors in *Platynereis*. One is the MIP GPCR, a sex-peptide-receptor ortholog, the other one is a peptide-gated channel. Peptide-gated channels are also present in the hydrozoan *Hydra*. The authors should rewrite this part of the Discussion section. In light of these studies, the authors may also want to discuss the potential nature of the *Nematostella* GLWamide receptor(s).

9) The raw data in Figure 5 should be supplied. In addition, the authors should replete the data to show the spread of the data points.

Some raw confocal image stack of the GLWamide in situs and immunos should be uploaded as source data files.

10) One final shared concern is with the interpretation of staining results, where it is suggested that GLWamide is expressed in sensory neurons. It is customary in the *Nematostella* community to refer to some neurons as sensory based on their position and potential similarity to putative cnidarian sensory cell types listed in the literature. However, we are not aware of any direct physiological or genetic evidence indicating that these cells play a sensory role in *Nematostella*, or what senses they may underlie. Furthermore, it is not known if such cells have non-sensory functions in neuronal signaling. It might, therefore, be best to say these cells are putative sensory cells or could be sensory cells and discuss the evidence for that in more detail. If there is indeed solid evidence, then perhaps that should be taken into account in the discussion: does the presence of GLWamide in specific sensory cells link specific environmental cues to polyp morphogenesis?

---

## [Author Response]

Essential revisions:Because of the importance of this study, we would like for the authors to address the following issues:1) The results that were obtained from the experiments reported are interesting; however, the authors' conclusion is based on the assumption that the gene in question is the only GLWamide-encoding gene in Nematostella. What is the evidence that no other related peptide is present (i.e., very short seq: LWG followed by K or R; easy to overlook), given also that the current genome assembly isn't great (see Figure 5—figure supplement 1)?

The lack of anti-GLWamide antibody staining in *glw* null mutants (Figure 5—figure supplement 1) provides evidence that there is no other gene that encodes GLWamides. If there were other GLWamide-encoding genes, the anti-GLWamide antibody should show reactivity in *glw* mutants, which we did not observe. See also our response to reviewers’ comment 2 below.

2) The antibody they used was generated against CEAGAPGLWamide and would probably not detect other Wamide-like peptides. The delayed metamorphosis reported by the authors is also consistent with the presence of an additional XWamide/LWamide that may induce metamorphosis less effectively. Also, Nematostella metamorphosis is different from other studied cnidarians (hydrozoans and corals) in that it does not require an exogenous trigger. Hence, these findings could be true for Nematostella but not necessarily for other cnidarians, where a microbial film or LWamide change metamorphosis rates from <5% in their absence to >95% when applied. Our recommendation: either provide evidence to exclude other LWamide encoding genes in Nematostella or, adapt the conclusion to the actual findings.

We think that our antibody should be able to detect other GLWamide-like peptides, if present, for the following reasons. First, this is a polyclonal antibody, and therefore the reactivity is expected to occur not only with the CEAGAPGLWamide-specific region not found in other GLWamide peptides, but also with the conserved C-terminal region. Second, the antibody was affinity-absorbed with non-amidated peptides KECPPGLWGC-cooh and KECLPGVWG-cooh to select for the antibody that would specifically react with the C-terminally amidated form of GLW peptides. Third, we have used this antibody to immunostain *Aurelia* ephyrae, and obtained a staining pattern consistent with that obtained by using an antibody against EQPGLWamide (Nakanishi et al., 2009), indicating that our antibody can cross-react with GLWamides from a different cnidarian; in the revised version of the manuscript we have included a supplemental figure (Figure 2—figure supplement 1) demonstrating this. We are therefore relatively confident that our GLWamide antibody should cross-react with GLWamides – irrespective of their N-terminal sequence – in *Nematostella*.

3) The paper would benefit from a better discussion of previous work done on the role of LWamides in cnidarians. For example, McFarlane et al. (1987), Takahashi et al. (1997), and Katsukura et al. (2004) reported that LWamide peptides (also known in Hydra as hym-54) cause a rhythmic contraction in sea anemone and hydrozoan muscles (at lower concentrations). Has there been any defect in locomotion or phototaxis in KO animals?

We have added a brief summary of existing ideas about Wamide's role in regulating the behavior of animals in the Discussion section. We saw no obvious defects in locomotion or photic behavior in GLWamide knockout animals, but further investigation is required to ascertain this. We think that thorough behavioral analyses are beyond the scope of this paper.

4) The authors write that previous work suggested the Wamides are required for metamorphosis. This is not entirely correct, and the text should be carefully revised. Conzelmann et al. showed that exogenous MIP can 'induce' or 'trigger' larval settlement and that the effects on the cilia require the MIP-receptor. They did not claim that MIP is 'required' for settlement.

We have revised the text accordingly.

5) In the Discussion section, the authors refer to a study by Plickert et al. on the role of GLWamide peptides in regulating Hydractinia metamorphosis. Plickert et al. concluded that GLWamide was required for metamorphosis in Hydractinia. This was based on two lines of evidence. One was the RNAi knockdown of the proneuropeptide that led to a delay/reduced rate in metamorphosis. The other was the pharmacological inhibition of the α-amidating enzyme that is required to convert the C-terminal Gly into an amide group during neuropeptide maturation. Inhibition of this enzyme with the copper chelators diethyl-dithiocarbamate (DDC) or tetra-ethyl-thiuram-disulfide (TETD) completely blocked metamorphosis. This effect could be rescued by synthetic GLWamide. Nakanishi and Martindale only mention the RNAi results and argue that the data in the Plickert et al. paper are consistent with GLWamide having an accelerating role. However, they should also discuss the inhibitor results and the rescue by the peptide, which suggest a requirement of amidated peptides.

Accordingly, we have included the discussion of evidence from the pharmacological and rescue experiments in the Introduction, and suggested the possibility, in the Discussion section, that amidated peptides we did not examine in this paper may be necessary for metamorphosis in *Nematostella*.

6) It may well be that there are species-specific differences in the involvement of GLWamides in the regulation of metamorphosis. The discussion presents the role of GLWamides and MIPs in general, assuming a conserved role. It should be made clear when the authors talk about the function of the peptide in Nematostella and when they suggest a conservation of function in other species.

We have revised the text to make explicit which species are referred to when statements on peptide function are made.

7) The following two statements should be carefully reworded:Introduction: the authors write that "Contrary to the hypothesis that Wamide is necessary for life cycle transition, we report that Wamide knockout mutant larvae can transform into morphologically normal polyps."andResults section: "The above results are incompatible with the previously proposed hypothesis that GLWamide is an indispensable component of the regulatory system controlling cnidarian metamorphosis (e.g. Plickert et al., 2003)."This work was done in Nematostella and the hypothesis that Wamide is necessary for lifecycle transition was proposed based on experiments in Hydractinia. The two diverged in or prior to the Cambrian. See also previous comments about the copper chelator experiments in Hydractinia.

We have revised the text to make explicit the species being discussed for the first statement and removed the second statement.

8) The authors write that "in the annelid polychaete Platynereis, MIP (Wamide) receptor expression is necessary for larval settlement behavior (Conzelmann et al., 2013).… reducing the expression of MIP by morpholino antisense oligonucleotide inhibits feeding behavior in the juvenile, but not metamorphosis (Williams, Conzelmann, and Jekely, 2015). This suggests the presence of as-yet-unknown ligands for the "MIP" receptor." The conclusion that the Platynereis MIP receptor has other ligands is not supported by the available data. First, the morpholino-mediated knockdown of the MIP receptor by Conzelmann et al. only abrogated the effect of MIP on ciliary closures. A longer-term effect on spontaneous settlement behavior (i.e. without extra MIP addition) was not investigated. So, the statement "receptor expression is necessary for larval settlement" is not precise.

We have clarified that MIP receptor expression is necessary for *MIP-induced* settlement behavior.

Likewise, the observation that MIP knockdown by morpholinos inhibits feeding behavior in the juvenile, but not metamorphosis, does not mean that the MIP receptor has other ligands.

Indeed, this is only one of several possibilities to account for these data from *Platynereis*. Another is, as mentioned in the text, incomplete knockdown of MIP so that the residual MIPs were sufficient to induce settlement behavior through the MIP receptor. It is also possible that MIP does not endogenously control larval settlement and metamorphosis. Yet another possibility is that metamorphosis is decoupled from larval settlement so that MIP-regulated settlement behavior is not required for metamorphosis. These additional possibilities are now discussed in the revised text.

Instead, what has been shown recently, is that MIP peptides signal through two different receptors in Platynereis. One is the MIP GPCR, a sex-peptide-receptor ortholog, the other one is a peptide-gated channel. Peptide-gated channels are also present in the hydrozoan Hydra. The authors should rewrite this part of the Discussion section. In light of these studies, the authors may also want to discuss the potential nature of the Nematostella GLWamide receptor(s).

It is not clear to us how having an additional MIP *receptor* would explain the apparent lack of defects in life cycle transition of MIP morphants. We avoided speculating on the nature of *Nematostella* GLWamide receptors, but have included a suggestion in Discussion section that it is one important aspect of Wamide-dependent regulation of developmental timing that should be investigated in the future.

9) The raw data in Figure 5 should be supplied. In addition, the authors should replete the data to show the spread of the data points.

We have revised Figure 5 and the source data file for Figure 5 as suggested.

Some raw confocal image stack of the GLWamide in situs and immunos should be uploaded as source data files.

We have uploaded raw confocal image stacks of the GLWamide immunostaining and in situ data as source data files.

10) One final shared concern is with the interpretation of staining results, where it is suggested that GLWamide is expressed in sensory neurons. It is customary in the Nematostella community to refer to some neurons as sensory based on their position and potential similarity to putative cnidarian sensory cell types listed in the literature. However, we are not aware of any direct physiological or genetic evidence indicating that these cells play a sensory role in Nematostella, or what senses they may underlie. Furthermore, it is not known if such cells have non-sensory functions in neuronal signaling. It might, therefore, be best to say these cells are putative sensory cells or could be sensory cells and discuss the evidence for that in more detail. If there is indeed solid evidence, then perhaps that should be taken into account in the discussion: does the presence of GLWamide in specific sensory cells link specific environmental cues to polyp morphogenesis?

The reviewer is correct – there is no physiological or genetic evidence that any of the GLWamide-expressing cells are sensory. However, this is the case for the majority of cnidarian ‘sensory cells’ (except in rare cases such as an opsin-expressing photosensory cell of a cubozoan). In cnidarian neurobiology literature beyond that concerning *Nematostella*, the definition of a ‘sensory cell’ is primarily morphological – a bipolar epidermal cell that is oriented perpendicular to the mesoglea and has an apical cilium exposed to the external environment, as well as basal thin neurite extensions (e.g. Thomas and Edwards, 1991). We consider it appropriate to follow the convention of cnidarian literature given that *Nematostella* is a cnidarian.